# Genetic Diversity, Population Structure, and Environmental Adaptation Signatures of Chinese Coastal Hard-Shell Mussel *Mytilus coruscus* Revealed by Whole-Genome Sequencing

**DOI:** 10.3390/ijms241713641

**Published:** 2023-09-04

**Authors:** Feng Guo, Yingying Ye, Kecheng Zhu, Shuangrui Lin, Yuxia Wang, Zhenyu Dong, Ronghui Yao, Hongfei Li, Weifeng Wang, Zhi Liao, Baoying Guo, Xiaojun Yan

**Affiliations:** 1National Engineering Research Center for Marine Aquaculture, Zhejiang Ocean University, Zhoushan 316022, China; agsguo@163.com (F.G.); yeyy@zjou.edu.cn (Y.Y.); citric-acid@foxmail.com (S.L.); lhf612@126.com (H.L.); 2Key Laboratory of South China Sea Fishery Resources Exploitation and Utilization, Ministry of Agriculture and Rural Affairs, South China Sea Fisheries Research Institute, Chinese Academy of Fishery Sciences, Guangzhou 510300, China; zhukecheng@scsfri.ac.cn; 3Marine Science and Technology College, Zhejiang Ocean University, Zhoushan 316022, China; wangyuxia97@163.com (Y.W.); dongzhenyu2021@163.com (Z.D.); 17662452562@163.com (R.Y.); wangwf@zjou.edu.cn (W.W.); liaozhi@zjou.edu.cn (Z.L.)

**Keywords:** *Mytilus coruscus*, whole-genome resequencing, single nucleotide polymorphisms, genetic diversity, selection signatures

## Abstract

The hard-shell mussel (*Mytilus coruscus*) is widespread in the temperate coastal areas of the northwest Pacific and holds a significant position in the shellfish aquaculture market in China. However, the natural resources of this species have been declining, and population genetic studies of *M. coruscus* are also lacking. In this study, we conducted whole-genome resequencing (WGR) of *M. coruscus* from eight different latitudes along the Chinese coast and identified a total of 25,859,986 single nucleotide polymorphism (SNP) markers. Our findings indicated that the genetic diversity of *M. coruscus* from the Zhoushan region was lower compared with populations from other regions. Furthermore, we observed that the evolutionary tree clustered into two primary branches, and the Zhangzhou (ZZ) population was in a separate branch. The ZZ population was partly isolated from populations in other regions, but the distribution of branches was not geographically homogeneous, and a nested pattern emerged, consistent with the population differentiation index (*F*_ST_) results. To investigate the selection characteristics, we utilized the northern *M. coruscus* populations (Dalian and Qingdao) and the central populations (Zhoushan and Xiangshan) as reference populations and the southern ZZ population as the target population. Our selection scan analysis identified several genes associated with thermal responses, including *Hsp70* and *CYP450*. These genes may play important roles in the adaptation of *M. coruscus* to different living environments. Overall, our study provides a comprehensive understanding of the genomic diversity of coastal *M. coruscus* in China and is a valuable resource for future studies on genetic breeding and the evolutionary adaptation of this species.

## 1. Introduction

*Mytilus coruscus*, commonly known as the hard-shelled mussel, has a wide distribution area in the northwest Pacific Ocean, including Hokkaido, Japan; Jeju Island, Korea; the Yellow Sea; the Bohai Sea; the East China Sea; and other regions [1,2]. Not only does this warm-water marine shellfish play a crucial role in maintaining the ecological balance between intertidal and subtidal communities, but it also has an important economic value as one of the important sources of edible shellfish in some Asian countries [3,4]. *M. coruscus* is the primary mussel species cultured in Zhejiang province and is a crucial species for marine aquaculture in China. In 2021, the mussel culture area in Zhejiang was 1837 hectares, with a production of 227,700 tons, ranking first in the country. Shengsi Islands, located in Zhoushan City, are the main production area for *M. coruscus* in Zhejiang (source: https://www.zj.gov.cn/ (accessed on 1 March 2023)). In recent decades, there has been a decline in the natural juvenile stocks of *M. coruscus*. Surprisingly, despite this decline, the number of mussel farms has increased, with a dependency on natural populations. Furthermore, the rapid depletion of the *M. coruscus* resource can be attributed to the overexploitation of fishery resources, severe pollution in offshore areas, and ecological degradation [5,6]. Modern genetics suggest that the genetic diversity of a species is closely related to its adaptive capacity, survival ability, and evolutionary potential [7]. Research to evaluate the level of genetic diversity of species germplasm resources is a prerequisite and basis for the recovery and sustainable use of natural resources. Therefore, it is necessary to research the genetic diversity of *M. coruscus* and to protect the resources of this species.

To date, several studies have been conducted on *M. coruscus*, focusing on its physiological and biochemical functions, biomaterials, adaptation to climate change, bacterial-host interactions, and biofouling and antifouling properties. It serves as a model organism for these research areas and plays a crucial role in the fields of economy, ecology, and biology [8,9,10,11,12,13,14,15]. However, studies on its population genetics, particularly large-scale population genetic studies, have been scarce. Previous studies have been limited to specific areas or seas, using a limited number of molecular markers. For example, Ye et al. (2013) sequenced and analyzed the mitochondrial DNA gene sequences of six regional populations of *M. coruscus* in the East China Sea and mitochondrial 16SrRNAs of *M. coruscus* from four regions along the southeast coast [16,17,18,19]. Similarly, Yi et al. (2021) used mitochondrial cytochrome oxidase-based gene sequences to explore the genetic diversity and population structure of *M. coruscus* species in Korea [20]. These studies provided insufficient information due to the low number of polymorphic markers and fewer molecular markers. Consequently, analyzing the entire genome sequence of *M. coruscus* through WGR is deemed to provide more comprehensive and in-depth information on the population genetic relationships of *M. coruscus*.

In recent years, next-generation sequencing (NGS) technologies have facilitated the discovery and acquisition of SNPs, which has enabled more accurate and efficient population genetic studies at the genomic level [21,22]. Whole genome sequencing (WGS) technology has been extensively used in genomic studies of aquaculture species to investigate evolutionary processes, population structure, environmental adaptation, and natural or artificial selection using SNPs, as well as to identify candidate genes associated with important traits [23]. For example, Wu et al. (2018) assembled the chromosomal-level genome of a female *Crassostrea ariakensis* and sequenced the genomes of 261 individuals from three typical habitats in China to examine the phylogenetic relationships among the three geographic populations and to detect selective traits of temperature and salinity adaptation in the southern and northern populations [24]. Lv et al. (2019) studied the population structure and genetic diversity of two farmed populations (HD and ZH) and one natural population (DL) of *Mizuhopecten yessoensis* from Dalian Zhangzi Island by analyzing whole genome resequencing data, and identified candidate genes associated with the phenotype of the dominant population, which laid the foundation for exploring scallop breeding [25]. In the context of mussel genome research, based on the data publicly available in the NCBI database, a total of 20 genome sequences of Mytilidae species have been disclosed. These primarily encompass *Mytilus galloprovincialis*, *M. edulis*, *M. californianus*, *M. chilensis*, and *Mytilisepta virgata*. Lallias et al. (2007) initially constructed a linkage map for *M. edulis* [26], while employing a low-density SNP panel to delineate species and genetic lineages within the *Mytilus* genus [27,28]; investigated the correlations between shell traits and species [29]; and reconstructed lineages [30]. Additionally, Yang et al. (2021) utilized Oxford Nanopore Technologies sequencing, Illumina sequencing, and high-throughput chromosome conformation capture techniques to establish the chromosome-level genome assembly for *M. coruscus* [31]. This furnished a foundational platform for studying the genetic diversity and adaptability of this species. According to Sun et al. (2017), significant close relationships exist between *M. galloprovincialis* and *M. edulis* and *M. coruscus*, as well as the latter two species, especially between *M. galloprovincialis* and *M. edulis* [32]. Corrochano-Fraile et al. (2022) used the closely related species *M. coruscus* and *M. galloprovincialis* for genome assessment, completed the assembly of the *M. edulis* genome, and performed a positive selection gene identification and functional analysis by calculating the evolutionary rate of three species in the family Mytilidae (*M. edulis* and *M. galloprovincialis*, *M. edulis* and *M. coruscus*, and *M. galloprovincialis* and *M. coruscus*) for the identification and functional analysis of positively selected genes [33].

In this study, we aimed to deepen the understanding of *M. coruscus* genetics by re-sequencing the entire genomes of 80 mussels from eight distinct regions along the Chinese coast. Our objectives were to analyze the genetic diversity and population structure of *M. coruscus* from different regions, identify the genomic selection signatures among these groups, and investigate their habitat adaptation. Our findings will supplement the current *M. coruscus* genomic resources and provide a foundation for future conservation of genetic diversity and the improvement of cultured *M. coruscus* species.

## 2. Results

### 2.1. Resequencing and Variant Discovery

We performed whole genome resequencing of 80 *M. coruscus* individuals from eight sites and obtained a total of 6.96 billion high-quality paired-end reads, with an average data size of 56 GB per sample and an average sequencing depth of 14× (Appendix A). The average mapping rate of clean reads to the reference genome was 96.3%, and a total of 269,960,788 SNPs were initially identified. After multi-step filtering of the variant detection data, a total of 25,859,986 high-quality SNPs were retained for all of the individuals (Table 1). These SNPs were distributed across the 14 chromosomes of the *M. coruscus* genome, with 22,834,368 (88.3%) on the chromosomes and 3,025,618 (11.7%) on unplaced scaffolds. The majority of SNPs were evenly distributed across the 14 chromosomes of *M. coruscus*, with roughly 13 SNPs per Kb (Figure 1a). An analysis of these single nucleotide substitution types indicated that of the 22,834,368 SNPs, 12,162,281 (53.26%) were categorized as transitions (A/G and C/T) and 10,672,087 (46.74%) were categorized as transversions (C/G, A/T, G/T, and A/C). The transitions (Ts) to transversions (Tv) ratio was 1.14. Within transitions, the number of A/G types and C/T types were almost equal in size, with 6,085,223 and 6,077,058, respectively. Among the transversions, A/T types were the most common, accounting for 42.91% (4,579,557) of the total transversions. Two types, A/C and G/T, represented almost the same percentage, with 23.62% (2,521,229) and 23.68% (2,526,829), respectively. Finally, the smallest share was the C/G type, representing only 9.79% (1,044,472) of all substitutions (Figure 1b).

### 2.2. SNPs Annotation and Location Distribution

We conducted genome-wide variant structure annotation using ANNOVAR software and found that 52.1% (11,888,903) of SNPs were located in intergenic, upstream, and downstream regions of the genes, while the other 47.9% (10,930,488) were distributed within genes, including different types of exons, introns, Stopgain, Stoploss, unknown, and SNV. Of all the genetic region variant loci, most were distributed in the intron region, accounting for 95.7% (21,838,158) of the total number of SNPs. Only 4.2% (966,629) of the SNPs were located in exonic regions, with synonymous substitutions and nonsynonymous substitutions accounting for 1.94% (447,427) and 1.77% (490,247) of the total SNPs, respectively (Table 2 and Figure 1c).

### 2.3. Population Genetic Diversity and Linkage Disequilibrium

To gain further insight into the genetic diversity of *M. coruscus* populations in eight regions, we calculated several metrics including genome-wide nucleotide diversity (π), inbreeding coefficient (*F*_IS_), observed heterozygosity (Ho), and expected heterozygosity (He). Our analysis revealed that the TS and ZZ populations had the highest genetic diversity based on observed heterozygosity (Ho = 0.2831, 0.2832) and expected heterozygosity (He = 0.2807, 0.2808). All of the populations showed moderately high nucleotide diversity, with the TS, XS, and ZZ populations exhibiting the highest nucleotide diversity (π = 3.793 × 10^−3^, 3.788 × 10^−3^, 3.787 × 10^−3^), and the ZS population showing the lowest nucleotide diversity (π = 3.213 × 10^−3^). It is worth noting that nucleotide diversity is a reflection of genetic diversity, to a certain extent. Inbreeding levels varied among the populations, with the YH and DL populations displaying higher levels compared with the other populations. All populations passed the HWE test, and the mean *p*-values of HWE ranged from 0.810 to 0.843 (Table 3 and Appendix A). Our analysis of linkage disequilibrium (LD) indicated that the decay rate of LD coefficients varied across populations. The TS, ZZ, YH, and XS populations showed a more rapid decay rate and lower levels of LD, while the QD, LYG, and DL populations exhibited a slightly slower decay rate than the former four populations. However, the ZS group showed the slowest decline rate in LD (Figure 1d).

### 2.4. Genetic Population Structure

We analyzed the genetic relationships and differences among populations using 2,883,362 LD-filtered SNPs. The results of the principal component analysis (PCA) showed that individuals from each group were closely clustered together, but they were not clearly separated from other populations. Only a few samples from the ZZ region and the LYG region showed significant deviation from other district groups, indicating a high degree of homogeneity among *M. coruscus* populations (Figure 2a). The NJ tree also revealed a scattered clustering pattern among samples that did not follow the characteristics of samples or geographical distribution, consistent with the results of the PCA analysis (Figure 2b).

To further investigate the relatedness between populations, we performed an unsupervised clustering analysis with ADMIXTURE. The cross-validation (CV) error was at its lowest at k = 1, indicating the presence of one ancestral cluster. At k = 2, a red cluster with a score above 0.75 consisted of 15 individuals, 8 from ZZ, 3 from LYG, 2 from Zhoushan, and 1 each from YH and TS. Among the 80 individuals, 53 showed mixed ancestry. Under K = 3 and 4, all eight populations were a mixture of the same three genetic components, suggesting a common ancestral origin. However, the LYG and ZZ populations appeared to show weak differentiation (Figure 2c and Appendix A).

### 2.5. Trends in Historical Effective Population Size

The neutrality test results indicated positive values for all eight regions on the Chinese coast, suggesting that the *M. coruscus* population was in a state of population contraction in all of them. Additionally, the results of SMC++ analysis revealed that the effective populations of *M. coruscus* in all eight regions have undergone a continuous decline for millions of years. Specifically, the Ne of the eight populations began to diverge at around 0.6 Ma, and then stabilized at 0.05 Ma (Figure 3 and Appendix A).

### 2.6. Exploration of Population Selection Signatures and GO Enrichment Analysis

To investigate how *M. coruscus* adapts to different environments in different geographic regions, we referred to an integrated selection method that combines *F*_ST_ and π-score ratios between paired groups [34], which has been used for whole-genome resequencing analyses of bivalve shellfish such as oysters [35] and scallops [25]. Specifically, the samples were divided into three groups based on geographic latitude, and genes were identified in the selected southern group (ZZ) using the northern group (DL and QD) and the central group (ZS and XS) as the reference group and the southern group (ZZ) as the target group. By filtering the regions with the largest differences in θπ ratios (northern θπ/southern θπ and central θπ/southern θπ), we considered the intersections of the top 5% θπ ratios and the top 5% gene groups with fixed index (*F*_ST_) values to be in the selection. We then combined overlapping windows for further analysis. Our results showed significant selection signals in the genes from the overlapping regions identified by both techniques. We also examined the degree of genome-wide population differentiation through Manhattan fixation index plots (*F*_ST_) and found that chromosomes 1, 2, and 4 of the ZZ population exhibited more significant differentiation features. This indicates that these regions experienced stronger selection, with 36.9% and 37.8% of the selection signatures coming from these chromosomes for the northern vs. southern and central vs. southern groups, respectively (Figure 4a and Figure 5a).

In the Zhangzhou population, a total of 896 genes were identified as being under selection pressure. Out of these, 232 genes (approximately 51.8%) were found to be shared in the comparison with both groups (central and northern), while 432 genes (approximately 48.2%) were found only in the comparison with one group (201 in the central group and 231 in the northern group) (Appendix A).

In contrast with the central group, we found 993 regions that were selectively scanned in the southern group, which contained 433 genes. These candidate genes were found to be associated with ion binding and multiple metabolic processes, including the lysophosphatidylglycerol acyltransferase 1 (*LPGAT1*). Using GO enrichment analysis, we identified 143 genes that were significantly enriched in 29 GO terms (*p*-value < 0.05). The most enriched GO terms were involved in the establishment or maintenance of apical/basal cell polarity (GO:0061245) and in the establishment or maintenance of bipolar cell polarity (GO:0061245) in the biological processes subclass. The most abundant cellular component categories were the apical part of the cell (GO:0045177), synaptic membrane (GO:0061245), and bipolar cell polarity (GO:0061245). The main molecular functions were ubiquitin-like protein transferase activity (GO:0019787) and catalytic activity, acting on RNA (GO:0140098) (Appendix A, Figure 4b,c).

Compared with the northern group, we identified 1095 selectively scanned regions in the southern group containing 463 genes showing selection signals. We found several genes associated with heat response and immune response, such as heat shock protein 70 (*Hsp70*), phosphoenolpyruvate carboxykinase (*PEPCK*), toll-like receptor 2 (*TLR2*), and ubiquitin carboxyl-terminal hydrolase 19 (*USP19*). Our GO enrichment analysis revealed that 41 genes were significantly enriched in 25 GO terms (*p*-value < 0.05), with the most enriched GO terms involved in the positive regulation of the pattern recognition receptor signaling pathway for immune system development (GO:0062208) and the positive regulation of interleukin-8 production (GO: 0032757), among others. The major molecular function represented was the extracellular matrix structural constituent (GO:0005201) (Appendix A, Figure 5b,c). These abundant genes could potentially play a significant role in the adaptation of *M. coruscus* to northern and southern environments.

## 3. Discussion

### 3.1. Genetic Diversity and Population Structure

Genetic diversity refers to the overall genetic composition of a species and is crucial for a population’s ability to adapt to environmental changes [36,37]. We searched the literature on bivalve nucleotide diversity and found that most bivalve species have only mitochondrial-tagged nucleotide diversity calculations and lack genome-wide nucleotide diversity data in bivalves (Appendix A). By comparing the same levels, we found that *M. coruscus* has higher nucleotide diversity at the whole-genome level than *M. yessoensis* and *C. ariakensis* and lower nucleotide diversity than *C. gigas* and *Semibalanus balanoides*, so we believe that *M. coruscus* may have a moderately high nucleotide diversity among bivalve shellfish [24,25,38,39]. Considering nucleotide diversity (π) as an indicator of genetic diversity magnitude among species [40], we propose that the genetic diversity of *M. coruscus* among bivalve species is also likely to be intermediate. However, there was not much variation in genetic diversity between regions, except for ZS, where genetic diversity was low (Appendix A). This could be attributed to aquaculture activities. In general, mass-cultured strains are believed to have lower population genetic diversity than wild populations due to the small population size and inbreeding decline [41,42]. Furthermore, the artificial culture environment is mixed with the natural habitat, leading to the introduction of a large number of cultured mussel larvae into natural waters, thus diluting the genetic diversity of natural populations [43]. To safeguard the genetic resources of relevant germplasm, the establishment of corresponding genetic conservation units may be an effective approach. Breeders should also be educated and encouraged to isolate breeding sites from the natural environment to minimize the flow of genetic information from breeding individuals to the natural environment.

Marine ecosystems generally have fewer barriers to dispersal than terrestrial or freshwater ecosystems [44,45,46]. Despite panmixia being rare in nature, many species, including plants and marine organisms, exhibit long-distance dispersal [47,48,49]. This leads to high levels of gene flow and low levels of genetic differentiation within species, due to high rates of dispersal between breeding sites [50]. For example, genetic panmixing is observed in marine invertebrates with large populations and a high dispersal potential [51]. The genetic structure analysis in this study showed that only the ZZ region exhibited genetic differentiation from the other seven regions along the Chinese coast. In addition, a high degree of genetic exchange was observed among all of the sampling sites, and there was no significant correlation between *F*_ST_ and geographic distance (Appendix A). This lack of geographic structure suggests a homogenization phenomenon in these populations. The robust genetic connectivity also indicates that these mussels could spread far along ocean currents, shaping their genetic structure.

Some organisms in the ocean release planktonic larvae that disperse using ocean currents over days to months, exhibiting effective long-distance larval dispersal and thus becoming the main source of their dispersal ability, such as *Chthamalus challengeri*, *Lipophrys pholis,* and *Conger myriaster* [52,53,54]. Like many other marine invertebrates, *M. coruscus* also have a free-swimming larval stage and are able to persist for longer periods of time (more than one month) and they are able to attach to natural and artificial substrate rafts, which allows them to reach their final settlement site even in the late larval stages [55,56]. Thus, this high dispersal potential is expected to lead to a high gene flow between populations, resulting in large-scale genetic homogenization. Additionally, human-mediated dispersal, such as the transplantation of mussel fry for breeding purposes and the transport of ballast water from ships, may also contribute to genetic homogeneity between coastal populations in eastern and northern China.

The pattern of ocean currents near the Chinese coast is complex and variable. The water exchange and seasonal inversion of the surface ocean circulation are driven by the prevailing monsoon-driven southwestward flow in the summer and northeastward flow in the winter [57,58]. The East China Sea region is a major production area for *M. coruscus* in China, rich in natural germplasm resources, and many larvae are dispersed in the region during the annual winter breeding season (Figure 6). While coastal currents in the region flow mainly from north to south in winter, the Kuroshio and its tributary warm currents isolate the exchange between the marginal sea and the ocean from south to north at the periphery. The winter is also a dry period for coastal rivers, with a significant reduction of flushing water from the estuary into the marginal sea This further weakens the influence on coastal currents, and the presence of eddies within each sea area also promotes the circulation within the sea area [59]. These marine conditions in the study area are conducive to *M. coruscus* migration along the Chinese coast, preserving high levels of genetic linkage between geographically dispersed communities. Therefore, *M. coruscus* appear to have a uniform phylogeographic pattern rather than a discrete genetic lineage along the Chinese coast.

### 3.2. Historical Demography and Population Contraction

The effective population size of *M. coruscus* in the eight geographic areas has been observed to exhibit a significant downward trend, which is likely due to a series of a series of ice-interglacial oscillations [60] that occurred during the Pleistocene epoch. This era, spanning the last million years, was marked by the presence of ice age cyclones during the most recent 800 kyr and 100 kyr interglacial periods dominated by the Late Pleistocene. The environmental changes during this time had a significant impact on population dynamics, leading to population growth and decline. Low sea levels during the Pleistocene epoch expanded the temperature and salinity range in nearshore waters. As a result, many shallow marine species faced harmful biological niche pressures and habitat shrinkage, leading to mass extinctions and sharp declines in population size. *M. coruscus*, being an intertidal species, was particularly influenced by terrestrial and oceanic factors, making it more susceptible to severe environmental stress [61]. Therefore, these changes may have caused a significant reduction in the effective population size of *M. coruscus*.

### 3.3. Selection Signatures and Environmental Adaptation

As the ocean’s temperature, pH, and other stressors associated with climate change constantly fluctuate, it is highly probable that many marine species will need to adapt to these changes to avoid extinction. To do so, these species require sufficient genetic variation in tolerance, which would enable them to adapt to environmental changes [51,62]. Although the genetic structure of *M. coruscus* did not change significantly in most areas of central and northern coastal China, the Zhangzhou Region in the far south showed signs of adaptive differentiation. We hypothesize that these differences may be due to fluctuations in environmental temperature across different regions. Temperature is a key factor that drives adaptive divergence in marine invertebrates, especially mussels [63,64,65]. Remote sensing data reveal notable variations in monthly mean temperature and sea surface temperature ranges between the northern and southern sample sites from November 2020 to November 2021. Especially in the coldest winter months, the sea surface temperature in the south is about 15 °C higher than in the north. Throughout the year, the southern sample sites are situated in a warmer environment, whereas the northern sample sites undergo fluctuating cold and warm temperatures (Appendix A). Consequently, the observed genomic traces of adaptive selection may be linked to these temperature disparities.

To investigate the genetic differentiation of *M. coruscus* in the south and north, we conducted selection signal analysis to identify selection signature genes specific to *M*. *coruscus* in the south. Next, we enriched these genes with Gene Ontology (GO) annotations to gain insight into their biological functions and their potential roles in adapting to the southern environment. As expected, many genes known to be associated with heat response were identified in the southern population that were not found in comparison with the central population, such as heat shock protein 70 kDa (*Hsp70*), cytochrome P450 protein (*CYP450*), and E3 ubiquitin-protein ligase (*CBLL1*, *RNF213*, and *RNF38_44*, *RNF1_2*) [66,67,68,69,70]. Heat shock proteins act as heat stress proteins and are widely present in various organisms [67,71]. As protein chaperones, they can assist in restoring the structure of unfolded proteins or promote the degradation of non-functional proteins and stabilize cell membranes, thereby protecting the organism from environmental stresses such as high temperatures [72,73]. Cytochrome P450 proteins (*CYP450*) belong to evolutionarily conserved antioxidant proteins and are considered to be another general molecular chaperone [74,75]. The expression of *HSPs* and *CYP450* genes is significantly induced by heat stress in intertidal species [76,77]. For example, these highly expressed *HSPs* and *CYP450* genes are thought to be important for the survival of the snail, Echinolittorina radiata, in the harsh thermal environment of rocky shores [68]. In addition, E3 ubiquitin-protein ligases are the carboxyl terminus of Hsp70-interacting proteins that act as molecular links to build bridges between cellular protein folding and degradation, and can function with protein chaperones such as Hsp70 and Hsp90 [69,78,79,80]. Furthermore, previous studies have found that the offspring of those individuals from warmer climates are more heat tolerant than those from cooler regions, possibly due to warmth acclimation [81,82,83]. For example, northern (warmer) populations of *Heliocidaris erythrogramma* exhibit significantly higher thermal tolerance than southern (cooler) populations, making it possible for northern thermotolerant populations to migrate poleward with the East Australian Current [83]. Similar phenomena may occur in *Mytilus edulis* and *Littorina littorea* [84,85]. Therefore, we suggest that these genes play an important role in southern adaptation to warmer environments, and we speculate that southern *M. coruscus* are better adapted to thermal environments than northern individuals.

Our study has provided a rich genomic resource of selection signatures related to temperature adaptation in *M. coruscus* at different latitudes, which provides important information for further investigation of the functional significance of these genes in *M. coruscus* and their role in influencing temperature adaptation. However, the estimated genetic diversity may be biased due to the small sample size, which may also result in a less comprehensive and uncertain identification of genomic variation. Therefore, future studies should use larger sample sizes to obtain more precise information on genetic variation and to determine the causal relationships between these selected genes and the environment.

## 4. Materials and Methods

### 4.1. Sample Collection and DNA Extraction

A total of 80 wild *M. coruscus* were collected from eight sites in Dalian (DL), Qingdao (QD), Lianyungang (LYG), Zhoushan (ZS), Xiangshan (XS), Yuhuan (YH), Taishan (TS), and Zhangzhou (ZZ), China, from September 2020 to March 2021 (Figure 6 and Appendix A). We sampled adductor muscle tissue from all of the samples and stored them individually in 95% ethanol at −20 °C, and extracted the genomic DNA from each sample using standard phenol/chloroform extraction [86]. The concentration of DNA samples was measured using Qubit Fluorescence Quantification; the integrity of DNA samples was measured using 1% agarose gel electrophoresis. The samples that passed the test were used for library preparation.

### 4.2. Library Construction and Genome Sequencing

DNA samples were first tested for concentration and integrity, and only samples that passed this test were used for library preparation, and then genomic DNA wsa randomly fragmented by sonication using the Covaris method. Subsequently, genomic DNA fragments with an average length of 200 to 400 base pairs were isolated using the agencourt AMPure XP-Medium kit. The selected genomic DNA fragments were end-repaired and 3′ adenylated, and then ligated to the adenylated fragments. The ligated DNA product was amplified by PCR and purified using the agencourt AMPure XP-Medium kit. After thermal denaturation (95 °C, 1 min), the purified PCR product was cyclized using the ligated oligonucleotide sequence to yield single-stranded circular DNA (ssCir DNA). This ssCir DNA constituted the final library and was rigorously characterized by quality control procedures. Libraries meeting quality standards were sequenced using the MGISEQ-2000 sequencer. ssCir DNA molecules were circularly replicated to form a DNA nanoball (DNB) containing more than 300 replicate copies. These DNPs were then loaded onto patterned nano-arrays using advanced high-density DNA nano-chip technology. Ultimately, 150 base pair reads were generated from the other end of the ssCir DNA using the combinatorial probe anchor synthesis (cPAS) method.

### 4.3. Sequencing Data Mapping and Variant Detection

The raw data generated were filtered using the Trimmomatic tool v0.39 [87]. The raw reads weer processed into high-quality clean reads following four strict filtering criteria: (1) removing the adapter, (2) reads that were less than 50 bp in length after removing the adapter were discarded, (3) the removal of reads with a mean sequencing quality MQ < 20, and (4) removal of localized low-quality bases. After removing the adapter and low-quality reads, the clean reads were further checked for quality using FastQC v0.11.9 (http://www.bioinformatics.babraham.ac.uk/projects/fastqc/ (accessed on 11 May 2022)).

The sequencing results were quality controlled in the above steps and the reads were aligned to the *M. coruscus* reference genome (unpublished data) using the BWA-MEM algorithm of the Burrows−Wheeler Aligner software v0.7.17-r1188 [88]. We used SAMtools software v1.15 [89] to convert the mapping results into BAM file format and to perform filtering of low-quality mapping data (MQ < 20). Then, we employed Picard Tools software v1.119 (http://broadinstitute.github.io/picard/ (accessed on 1 June 2022)) to sort, organize, and eliminate PCR duplicates, and to generate an index of the BAM files. For variant calling, we employed GATK software v4.1.9.0 [90] and utilized the haplotype caller module [91]. Variant filtration was performed using GATK and the recommended standard filter for SNPs (“QD < 2.0||FS > 60.0||MQ < 40.0||MQRankSum < −12.5||ReadPosRankSum < −8.0”).

### 4.4. Variant Annotation

To annotate the location and other attributes of the variants, this study used ANNOVAR v2.4 [92] to determine their location and the characteristics in the genome. We used reference genome sequence information and genome annotation files of *M. coruscus* to classify all SNPs into exons, introns, upstream, downstream, and splice sites.

### 4.5. Genetic Diversity Analysis

The genetic diversity and population structure among varieties and cultivars, as well as the heterogeneity of each genetic parameter (He and Ho, respectively), minor allele frequencies (MAF), and nucleotide diversity (θπ) were estimated using PLINK (v1.9) [93] and VCFtools software v0.1.16 [94]. The interpopulation fixation index (*F*_ST_) values for all SNPs were calculated using the -fst option of the PLINK software. The Hardy−Weinberg balance was detected using the “-hew” option of PLINK. Calculation of the inbreeding number (*F*_IS_) in the population was performed using Genepop software v4.7 [95]. In addition, PopLDdecay v3.41 (https://github.com/BGI-shenzhen/PopLDdecay (accessed on 3 July 2022)) [96] was used to calculate the LD coefficient (r2) between two points in a series of sequences to estimate the LDdecay trend. The faster the decay of r^2^, the higher the genetic diversity of the population [97].

### 4.6. Population Structure Analysis and Historical Effective Population Size

The VCFtools software was utilized to convert the VCF files of 80 individual t *M. coruscus* into plink files. These files were then filtered based on a minimum allele frequency (MAF) of >0.01 to ensure that all populations contained at least three allelic mutations per locus. The genomic data were further processed using PLINK software to remove linked loci with the parameter “-indep-pairwise 50 5 0.2”. After filtering, a total of 2,883,362 SNPs were used for the principal component analysis (PCA), phylogenetic tree construction, and population structure analysis to illustrate the genetic evolutionary relationships among *M. coruscus* populations. Neighbor-Joining (NJ) trees were constructed by calculating the genetic distance matrix through PHYLIP v3.69 [98] software and then using this matrix through MEGA7 [99]. The population structure analysis was performed in the Admixture program v1.3.0 [100]. The number of cluster (K) values ranged from K = 1 to K = 4. The coefficient of variation (CV) error was used to determine the best K value.

Two strategies were used to extrapolate the historical demography. First, Tajima’s D [101] was calculated using VCFtools software for the neutrality test. This test was used to detect any deviations from the neutrality assumption, which would indicate recent population expansion or contraction. Secondly, we randomly selected four samples for each of the *M. coruscus* populations, resulting in a total of 32 samples. The SMC++ model v1.15.4 (https://github.com/popgenmethods/smcpp (accessed on 11 July 2022)) [102] was then used to estimate the historical effective population size for each of the selected populations separately. As there were no mutation rates available for *M. coruscus* or closely related species, we used a mutation transformation matrix based on oyster data [103]. A generation time of 1 year and a neutral mutation rate of µ = 1.39 × 10^−9^ were used [103,104]. The parameters were set to -c 10 -k 10 -m 1.39 × 10^−9^ -sp cubic.

### 4.7. Selective Signal Discovery and GO Enrichment

To compare population differences across geographic regions, we used a combination of the coefficient of fixation (*F*_ST_) and nucleotide diversity (π) metrics to detect possible selection traits [105]. The samples were first divided into three groups based on geographic latitude, with the northern (DL and QD) and central (ZS and XS) groups as the reference group and the southern (ZZ) group as the target group, and the genes were identified in the selected southern group [35]. A window size of 100 kb and a step size of 10 kb were used for assessing the selection features. The lower boundary of the diversity window was determined through a log10 transformation of the π values. This parameterization was discretized using an internal PERL script [106]. Candidate regions were identified based on the top 5% of the *F*_ST_ value to the ratio of π values [107]. Subsequently, potential candidate genes were extracted for the enrichment analysis.

To understand the potential functions and pathways of the candidate genes, we performed a Gene Ontology (GO) enrichment analysis using the R package clusterprofiler [108]. The false discovery rate (FDR) was adjusted for the *p*-value calculations, and an FDR threshold of less than or equal to 0.05 was used. A *p*-value < 0.05 was considered significant for the enrichment results.

## 5. Conclusions

In this study, we conducted whole-genome resequencing of *M. coruscus* from eight regions along the Chinese coast to investigate the genetic diversity and population structure. The findings revealed that the ZS region had a lower genetic diversity than the other regions, while the remaining seven regions showed no significant differences. The genetic structure of *M. coruscus* showed a trend of weakening, with partial differentiation observed in the ZZ region. Conversely, other regions did not show significant genetic differentiation, and there was no correlation between phylogenetic relationships and geographic distribution. Additionally, we assessed temperature adaptation in the southern and northern populations. These findings provide valuable insights into the genetic distribution of this marine organism and serve as a reference for genetic breeding and studies on the environmental adaptability of intertidal bivalves.

## Figures and Tables

**Figure 1 ijms-24-13641-f001:**
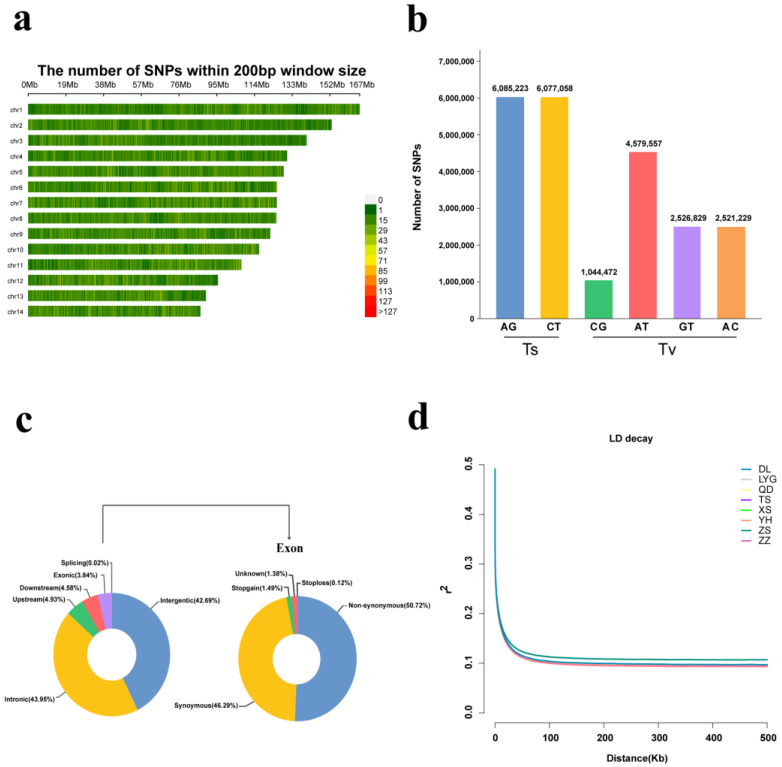
Variation information statistics and LD pattern. (**a**) Distribution of SNPs on 14 chromosomes in hard-shell mussels. Density results are displayed using the number of SNPs per 200 bp in the data expressed as a color index. (**b**) Transition and transversion of SNPs in the genome. (**c**) Location and number of SNPs in the gene structure. Number and proportion of SNPs probed in intergenomic regions and in the coding and non-coding sequences. In which SNP annotations in exons include synonymous and non-synonymous substitutions and their function is classified. (**d**) LD attenuation curves for eight sampling sites along the Chinese coast.

**Figure 2 ijms-24-13641-f002:**
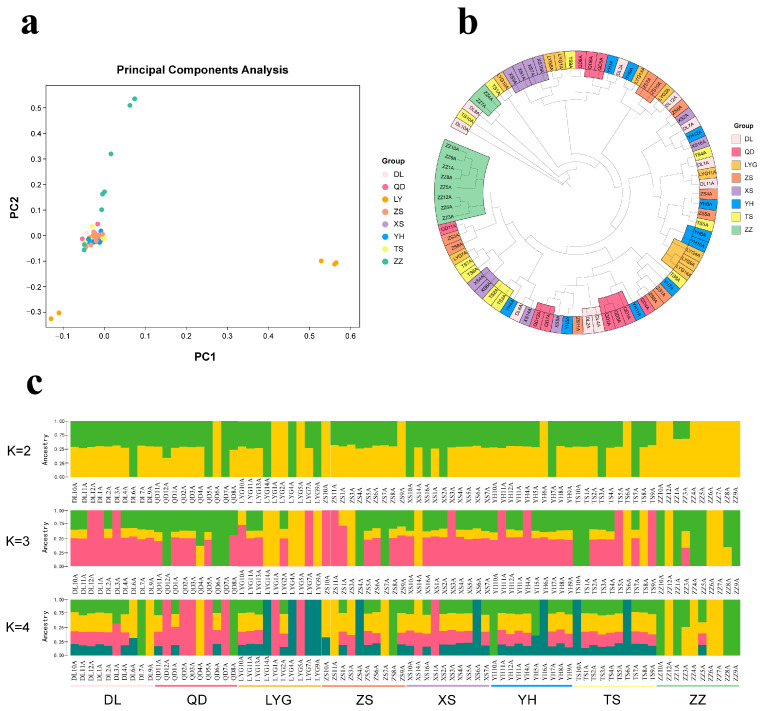
Population structure and phylogenetic analysis of hard-shell mussels from eight geographic locations. (**a**) Principal component analysis (PCA). (**b**) Neighbor-joining (NJ) phylogenetic tree. (**c**) Population structure clustering diagram.

**Figure 3 ijms-24-13641-f003:**
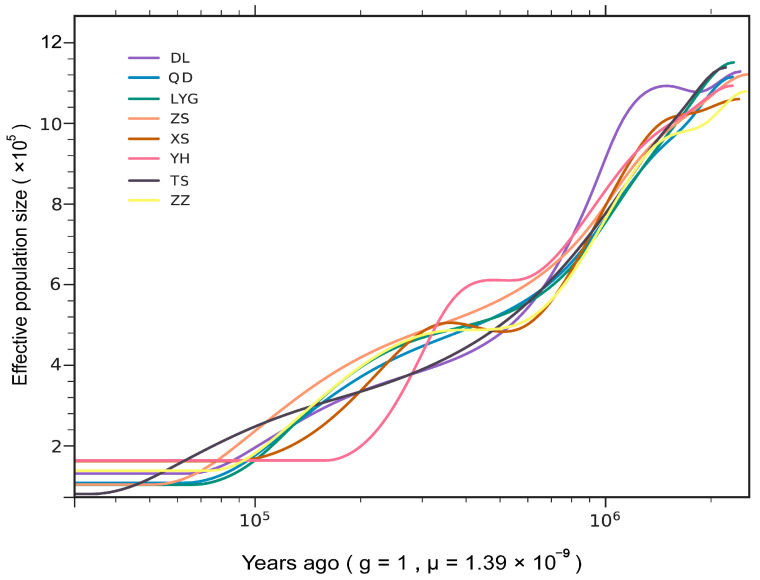
Results of the effective population size analysis of hard-shell mussels in eight coastal areas of China.

**Figure 4 ijms-24-13641-f004:**
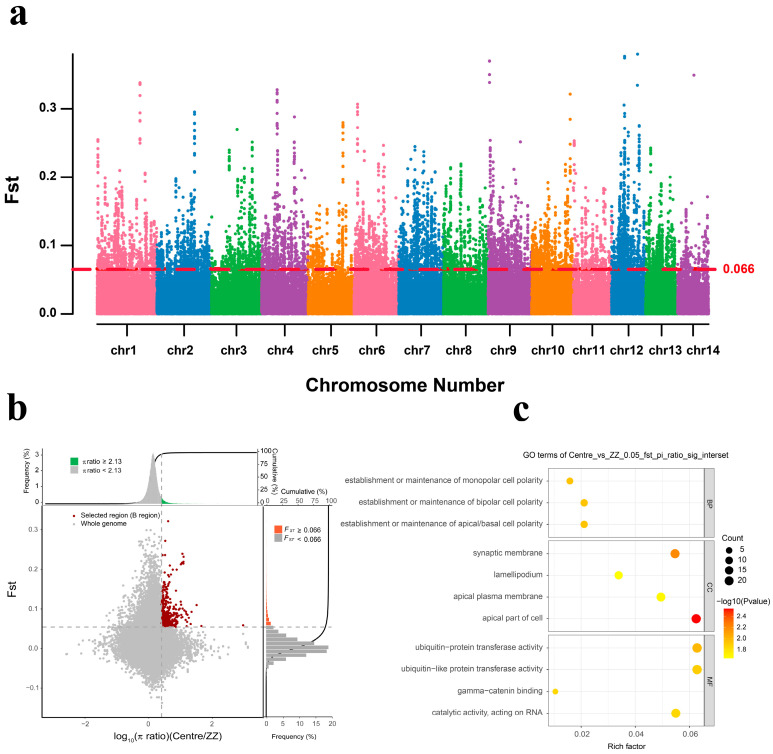
Candidate gene exploration and GO enrichment analysis of hard-shell mussels from the Zhangzhou Region (Centre_vs_ZZ). (**a**) Distribution of *F*_ST_ values on 14 chromosomes in the northern and Zhangzhou groups. The red dashed line indicates the significant threshold for identifying putative selection regions (top% 5 *F*_ST_ = 0.066, *p*-value < 0.05). (**b**) The central group is the control group and the Zhangzhou group is the selection group; 433 selection genes were screened. The red dots show the selective scan. (**c**) Results of GO enrichment analysis of the selected genes in Zhangzhou.

**Figure 5 ijms-24-13641-f005:**
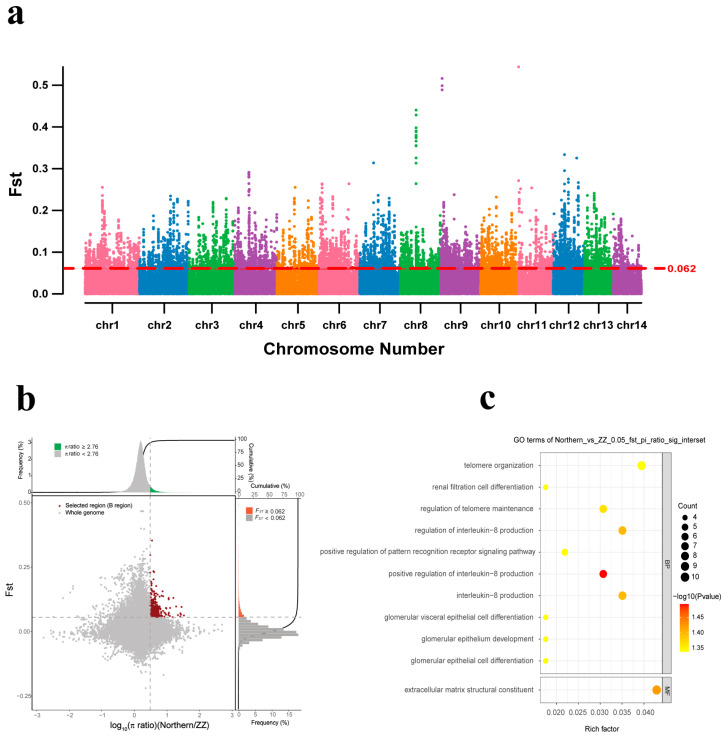
Candidate gene exploration and GO enrichment analysis of hard-shell mussels from the Zhangzhou Region (Northern_vs_ZZ). (**a**) Distribution of *F*_ST_ values on 14 chromosomes in the northern and Zhangzhou groups. The red dashed line indicates the significant threshold for identifying putative selection regions (top% 5 *F*_ST_ = 0.062, *p*-value < 0.05). (**b**) The northern group is the control group and the Zhangzhou group is the selection group; 433 selection genes were screened. The red dots show the selective scan. (**c**) Results of GO enrichment analysis of selected genes in Zhangzhou.

**Figure 6 ijms-24-13641-f006:**
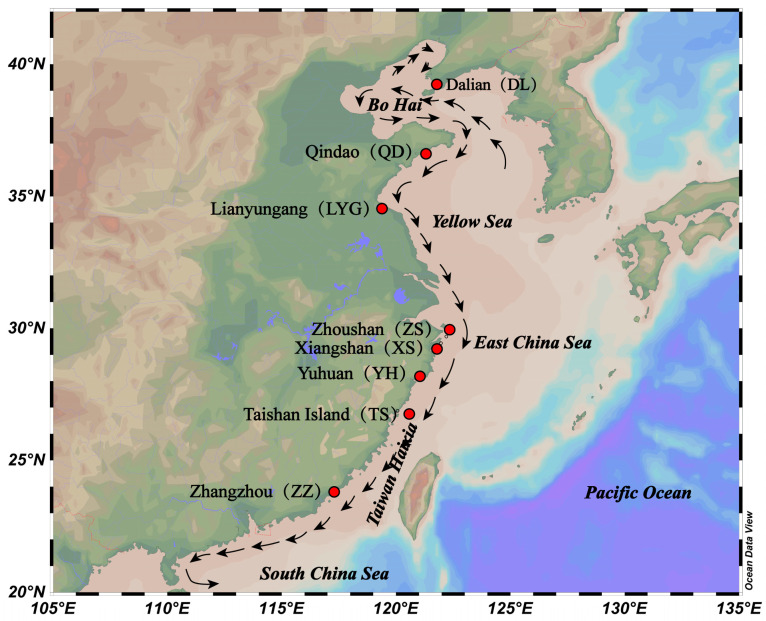
Geographical map of sampling sites for eight coastal populations in China. The alphabetical abbreviation of the location name of each population is defined in the text and the black arrows show winter currents off the coast of China.

**Table 1 ijms-24-13641-t001:** Number of variant loci after different filtering procedures.

Filtration Process Parameters	SNP Number
Total unfiltered SNPs	269,960,788
QD > 2, FS < 60, MQ > 40, MQRankSum > −12.5, ReadPosRankSum > −8.0	232,334,799
Minor allele frequency (MAF) ≥ 10%, <10% max-missing, SNPs with depth from 4 to 50×	25,859,986

**Table 2 ijms-24-13641-t002:** Classification of SNPs.

SNP Type		Count	Percent (%)
Upstream		1,241,961	4.92%
Intergentic		10,760,112	42.59%
Downstream		1,154,443	4.57%
Splicing		4029	0.02%
Intronic		11,078,046	43.85%
	Stopgain	14,406	0.06%
	Unknown	13,375	0.05%
Exonic	Stoploss	1174	0.004%
	Synonymous SNV	447,427	1.77%
	Non-synonymous SNV	490,247	1.94%

**Table 3 ijms-24-13641-t003:** Summary of the genetic diversity measurement for eight populations.

Statistic	Dalian (DL)	Qingdao (QD)	Lianyungang (LYG)	Zhoushan (ZS)	Xiangshan (XS)	Yuhuan (YH)	Taishan (TS)	Zhangzhou (ZZ)
Ho	0.277	0.278	0.281	0.282	0.281	0.275	0.283	0.283
He	0.281	0.281	0.281	0.281	0.281	0.281	0.281	0.281
π	3.770 × 10^−3^	3.667 × 10^−3^	3.781 × 10^−3^	3.213 × 10^−3^	3.788 × 10^−3^	3.727 × 10^−3^	3.793 × 10^−3^	3.787 × 10^−3^
Fis	0.014	0.011	0.0004	0.003	0.002	0.022	0.007	0.009
Tajima’sD	0.353	0.374	0.395	0.430	0.353	0.343	0.369	0.401
HWE *p*-value	0.809	0.813	0.813	0.843	0.814	0.810	0.814	0.816
MAF	0.194	0.187	0.188	0.189	0.188	0.186	0.188	0.188

He: expected heterozygosity; Ho: mean observed heterozygosity; π: population polymorphism index; Fis: inbreeding coefficient; HWE: *p*-value of Hardy−Weinberg equilibrium value (>0.05 is representative); MAF: minimum allele frequency; Tajima’s D: indicator of neutrality testing.

## Data Availability

The datasets generated and analyzed in this study are available in the NCBI SRA Repository. The Biological Project Accession Number is PRJNA972321.

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
