# Peer review of "Genetic Diversity, Population Structure, and Environmental Adaptation Signatures of Chinese Coastal Hard-Shell Mussel Mytilus coruscus Revealed by Whole-Genome Sequencing"

_ijms, 2023, doi:10.3390/ijms241713641_

Round 1

Reviewer 1 Report

This manuscript reports the results of whole genome resequencing on 80 individuals of Mytilus coruscus from 10 locations in China. The data are valuable, but parts of the manuscript are vague or unclear to me, and there are additional comparisons and analyses that could add much useful information with relatively little effort. I have the following suggestions for revision.

1) Whole-genome sequence data are available for Mytilus galloprovincialis, M. edulis, M. chiliensis, and M. californianus, yet there is no mention of any of these species in the manuscript. Which of these species is closest to M. coruscus? Is the divergence between them low enough that tests for selection that use the ratio of polymorphism to divergence could be used (tests that look for regions of low polymorphism/divergence as evidence of selective sweeps, McDonald-Kreitman test, etc.)?

2) This manuscript says the nucleotide diversity (pi) in M. coruscus is "high," but how does it compare to other marine bivalves with planktonic larvae? Is it notably higher, or lower, than most other bivalves? Or do most bivalves have similarly high values of pi, suggesting there may be an upper limit to nucleotide diversity? There is a substantial literature on levels of nucleotide diversity in marine organisms with planktonic larvae (and presumably very large population sizes) that the authors need to consider.

3) The manuscript says they used "a composite selection metric that combines FST and the ratio of π scores between pairwise groups." Their description of this method is very vague, both in the Methods section and the Results. Perhaps this is a standard method for detecting selection that I am unfamiliar with, and they just need to add a citation that explains it in more detail. Or perhaps it is a new method that the authors have developed. As it is now, I don't understand how they determined which genes were under selection.

4) Some of the other methods were described so vaguely that it would be impossible to replicate them. For example, section 4.2 says "After denaturing the PCR products to single-stranded, the cyclization reaction system was prepared, and the reaction was mixed thoroughly for a certain time at
appropriate temperature to obtain single-stranded cyclic products, and the uncycled linear products were eliminated." This would be very frustrating for anyone attempting to replicate the method. How were the PCR products denatured to single-stranded? How long is "a certain time," and what is the "appropriate temperature"? How were the uncycled linear products eliminated? There are many other examples in the Methods section of vague descriptions of methods.

The English is mostly fine, but there are some places that need editing. For example, section 4.2 says "Sequencing was mm performed by combined probe Probe-Anchor 419 Synthesis (cPAS) was sequenced to obtain pair-ended 150bp reads." The "mm" should be deleted, and I think "was sequenced" should be deleted. Section 4.3 says "Use SAMtools software to convert mapping results into BAM file format," it should say "We used SAMtools software..."

Reviewer 2 Report

The manuscript under review addresses several key aspects related to the genetic diversity, population structure, and environmental adaptation of the Chinese coastal hard-shell mussel, Mytilus coruscus. The study uses whole-genome resequencing data to uncover a substantial number of single nucleotide polymorphisms (SNPs) and applies various genetic analyses to explore the genetic diversity, population relationships, and adaptation mechanisms of this species. The study is both relevant and original, considering the importance of M. coruscus in the shellfish aquaculture market in China and the lack of comprehensive population genetic studies on this species.    The primary question addressed in this research is the genetic diversity, population structure, and adaptation mechanisms of M. coruscus along the Chinese coast. The study aims to provide a comprehensive understanding of the species' genomic diversity and to identify genetic traits associated with adaptation to different environmental conditions.    The topic of this study is highly relevant to the field, as it focuses on a commercially significant species, M. coruscus, whose natural resources have been declining. The study addresses a specific gap by conducting a population genetic analysis that covers a range of latitudinal locations along the Chinese coast. This geographic coverage, along with the application of whole-genome sequencing, adds originality to the research and contributes valuable insights to the field of population genetics.    This manuscript offers a significant contribution to the field by presenting a comprehensive analysis of the genetic diversity and population structure of M. coruscus in the context of environmental adaptation. The use of whole-genome resequencing provides a detailed genetic profile, and the identification of specific genes associated with thermal responses enhances our understanding of adaptation mechanisms.   Specific points   Methodology The methodology employed in this study is robust, particularly the use of whole-genome resequencing and the subsequent SNP analyses. However, there are a few areas where the authors could consider improvements: 
  1. Sample Size and Distribution: The rationale for the choice of the sample size and the distribution across different latitudes could be further explained. Consider expanding the sample size or providing additional justification for the chosen sample size to strengthen the representativeness of the genetic diversity. 
  2. Additional analyses, such as the assessment of potential biases in the SNP distribution, could provide deeper insights into the observed genetic patterns.
  3. Considering the potential importance of the findings, it would be beneficial to include or recommend further some form of validation, such as experimental validation of specific candidate genes associated with thermal responses, to reinforce the reliability of the results.
  Conclusion and discussion The conclusions drawn in the manuscript are largely consistent with the evidence and arguments presented. The study identifies clear genetic clusters, discusses population differentiation, and highlights specific genes associated with adaptation. However, the conclusion and discussion sections could be further emphasized by discussing the potential practical implications of the findings for the conservation and aquaculture management of M. coruscus.    In summary, this manuscript presents a valuable contribution to the field of population genetics, particularly in the context of genetic diversity, population structure, and adaptation mechanisms in M. coruscus. It addresses a relevant and specific gap, and the use of whole-genome resequencing is commendable. With some additional explanations and validation efforts, the manuscript has the potential to become an even more significant resource for researchers interested in genetic breeding, evolutionary adaptation, and conservation strategies for this economically important species.  

Minor points 

All over the manuscript, ‘M. coruscus’ should be italicized. 

Author Response

请参阅附件。

Round 2

Reviewer 1 Report

The authors have successfully addressed my comments on the first version, and the manuscript is greatly improved. The first paragraph of the Discussion says "We searched the literature on bivalve nucleotide diversity and found that most bivalve species have only mitochondrial-tagged nucleotide diversity calculations and lack genome-wide nucleotide diversity data in bivalves." The cover letter for the revised manuscript includes a table of nucleotide diversity values for bivalves, but it is not in the manuscript. I recommend that this table be included in the manuscript, but only the values for whole-genome diversity, not mitochondrial diversity. Also, the table and the text include Semibalanus balanoides, which is a barnacle (crustacean), not a mollusc. The authors should either remove Semibalanus, or expand the table and discussion to compare the diversity of all marine invertebrates with planktonic larvae, not just bivalve molluscs.
